Response inhibition and morphological awareness in children with attention deficit hyperactivity disorder: evidence from behavior and ERPs

Cheng Fang 1 2
Hu Xinhui 1 2
Chi Yawen 3
Yang Jie 1
Hu Changzhou 1
Wang Beini 1
Cui Jingjing 1
Wu Taoping 1 4
Chen Lixian 240434018@qq.com 5
Wang Rong 525250613@qq.com 1
1 Department of Psychiatry, Affiliated Kangning Hospital of Ningbo University , Ningbo , Zhejiang , China
2 Health Science Center, Ningbo University , Ningbo , Hubei , China
3 Institute of Education, China University of Geosciences , Wuhan , China
4 Department of Psychiatry, Seventh People’s Hospital , Cixi , Zhejiang , China
5 Department of Psychiatry, The Second People’s Hospital of Yuhuan , Yuhuan , Zhejiang , China
Caruana Nathan
Electronic publication date: 2025 Sep 16
Publication date: 2025
Volume: 13
Electronic Location ID: e19863
Received 2024 Oct 18; Accepted 2025 Jul 16
Copyright: ©2025 Cheng et al.
Copyright year: 2025
Copyright holder: Cheng et al.
License: This is an open access article distributed under the terms of the Creative Commons Attribution License, which permits unrestricted use, distribution, reproduction and adaptation in any medium and for any purpose provided that it is properly attributed. For attribution, the original author(s), title, publication source (PeerJ) and either DOI or URL of the article must be cited.
License URL: https://creativecommons.org/licenses/by/4.0/

Keywords: Attention deficit hyperactivity disorder, Event-related potentials, Response inhibition, Go/No-Go task, Morphological awareness

Funding: Zhejiang Medical and Health Science and Technology 2021KY330 Ningbo Natural Science Foundation 202003N4262 Ningbo Municipal Health and Health Scientific Research Project 2022Y21 Zhejiang Province Medical and Health Science and Technology Programme 2024KY558 Taizhou Social Development Science and Technology Programme 23ywb150 This work was supported by the Zhejiang Medical and Health Science and Technology (2021KY330), the Ningbo Natural Science Foundation (202003N4262), the Ningbo Municipal Health and Health Scientific Research Project(2022Y21), the Zhejiang Province Medical and Health Science and Technology Programme (2024KY558) and the Taizhou Social Development Science and Technology Programme (23ywb150). The funders had no role in study design, data collection and analysis, decision to publish, or preparation of the manuscript.

==============================
Background

Response inhibition is an important predictor of attention deficit hyperactivity disorder (ADHD), and many studies have shown that phonological awareness is associated with inhibition in native English-speaking children. Unlike English, which is phonetic, Chinese is an ideographic language. In the context of Chinese as a native language, do children with ADHD have deficits in morphological awareness? The present study explored the differences in response inhibition and morphological awareness between children with ADHD and typically developing (TD) children using behavioral data and event-related potentials (ERPs) to verify whether there is a morphological awareness deficit in children with ADHD.

Method

Go/No-go task was used to verify the presence of response inhibition deficits in children with ADHD, in which participants were required to respond rapidly to a “Go” stimulus and inhibit responses to a “No-go” stimulus. The Morphological Awareness task was used to verify the presence of a morphological awareness deficit in children with ADHD, in which participants were required to make judgement about true word and pseudo-word.

Results

Go/No-go task shows children with ADHD had significantly lower correct No-go stimuli and longer reaction times than TD children, lower evoked N200 amplitudes, and significantly impaired inhibitory control in children with ADHD. The morphological awareness task required participants to recognize pseudo-words and true words, and to respond to pseudo-words. The results showed no difference between the two groups of children in terms of correct response rate, N400 wave amplitude, and latency on the morphological awareness test.

Conclusion

The results of the study showed that children with ADHD have deficits in response inhibition compared to TD children and do not have significant deficits in morphological awareness.

Introduction

Attention deficit hyperactivity disorder (ADHD) is one of the most common neurodevelopmental disorders in children and adolescents with core symptoms of inattention, hyperactivity, and impulsivity disproportionate to the patient’s age level. The global prevalence of ADHD is estimated to be 7.2% (Thomas et al., 2015) and the domestic prevalence is 6.26% (The Subspecialty Group of Developmental and Behavioral Pediatrics, 2020). Some studies have shown that response inhibition is a major deficit in children with ADHD (Alexander, De Long & Strick, 1986; Senderecka et al., 2012), and response inhibition refers to the ability to inhibit responses to behaviors that are incongruent or conflicting with current goals (Chen, Rong & Li, 2013). Numerous studies have shown that children with ADHD exhibit significant behavioral deficits in inhibitory control tasks, such as elevated error rates and prolonged reaction times, accompanied by abnormal features of event-related potentials, such as prefrontal function-related reductions in N200 amplitude and prolonged P3 latencies (Pliszka, Liotti & Woldorff, 2000; Wang et al., 2005; Morand-Beaulieu et al., 2022). Research have shown (Pennington & Ozonoff, 1996; Orm et al., 2023) that inattention, hyperactivity, and impulsive behaviors in children with ADHD are closely related to deficits in response inhibition. Children’s attentional selection and inhibition abilities mature with age, but the development of response inhibition in children with ADHD typically lags behind that of typically developing (TD) children. Compared with TD children, children with ADHD have varying degrees of deficits in both response conflict and response arrest inhibitory functions compared to TD children (Wang, Wang & Zhou, 2006). Barkley (1997) proposed that deficits in inhibitory function are the core impairment in ADHD. His model describes three types of inhibition: inhibition of the prepotent response, stopping an ongoing response, and interference control. These inhibitory functions are typically measured using tasks such as Go/No-go task, Stop-Signal task, and the Stroop Color-Word test. The Go/No-go task assesses the ability to withhold a planned response, reflecting inhibitory control and conflict monitoring. In this task, participants are required to respond quickly to “Go” stimuli (e.g., specific letters, numbers, or colors) while inhibiting responses to “No-go” stimuli. Analyses of event-related potentials (ERPs) in children with ADHD during Go/No-go tasks have revealed impairments in the N200 component (200–300 ms), suggesting dysfunction in prefrontal cortical mechanisms underlying response inhibition (Baijot et al., 2017). A study comparing children with ADHD and those with reading disabilities (RD) found that children with ADHD exhibited reduced N200 amplitude over the right frontal lobe during Go/No-go tasks, as well as a lack of N200 modulation during successfully inhibited trials. In contrast, RD children showed no such abnormalities. These findings indicate that N200 abnormalities are specific markers of inhibitory deficits in ADHD and are linked to early stages of executive function processing (Liotti et al., 2010). Other scholars using the Stop-signal task reported that children with ADHD demonstrate greater N200 amplitudes and longer latencies compared to TD children (Senderecka et al., 2012).

Morphological awareness is an individual’s perception, manipulation and use of the smallest, meaningful linguistic units (morphemes) in language (Carlisle, 2000). Morphological awareness, as a core competence in language processing, involves the ability to perceive and manipulate the internal structure of words (e.g., roots and affixes), requiring individuals to break down and manipulate the morphological structure of words (e.g., the words “播放” in “播” and “放”), which is an important predictor of reading ability (Carlisle, 2000; Wu, Shu & Liu, 2005; Zhang et al., 2017). Several studies have found that inhibition is related to writing awareness, phonological awareness, word recognition, phonemic analysis, and early reading. Concurrent longitudinal studies of preschool and kindergarten children have found that inhibition predicts early literacy growth (Cameron et al., 2012; Miller et al., 2013). A study of native Spanish speakers showed significant correlations between inhibition and all indicators of vocabulary, phonological awareness, and knowledge of printed text (Lonigan et al., 2016). Several studies have shown that people with poor comprehension have problems suppressing irrelevant information, suggesting that poor comprehension is not only due to poor decoding skills, but may also be caused by poor inhibition (Borella, Carretti & Pelegrina, 2010; Gernsbacher & Faust, 1991; Yang et al., 2002; Wang & Zhu, 2009). As an ideograph, Chinese characters differ from epigraphic characters such as English. There are a large number of “one sound, many words” and “one word, many meanings” in Chinese. On the other hand, most words in Chinese are generally composed of two or more morphemes in a compound manner (Dong et al., 2014). Some researchers classify morphological awareness into homomorphic morpheme awareness, homophonic morpheme awareness and compound morpheme awareness (Liu et al., 2013). Homomorphic morpheme awareness is the ability to distinguish between Chinese characters with the same shape and different meanings. Homophonic morpheme awareness refers to the ability to distinguish between homophonic and different shaped Chinese characters, and they can help children distinguish the meaning and pronunciation of the same Chinese character in different contexts (Cheng, Wang & Wu, 2018). Studies of ERPs in TD children have found that abnormalities in morpheme processing can be reflected by difficulties in semantic integration of N400 components (Hill et al., 2002). Response inhibition as a core deficit in children with ADHD, so whether there is a deficit in morphological awareness in children with ADHD is yet to be proved by additional national studies. The present study intends to combine behavioral and electroencephalogram (EEG) data to investigate whether children with ADHD have deficits in morphological awareness in a Chinese-speaking language development system. In summary, this study proposes the hypothesis H(1): children with ADHD simultaneously exhibit deficits in response inhibition and morphological awareness.

Methods

Participants

Students in grades 3 to 5 of an elementary school in Zhejiang Province Ningbo city, who voluntarily participated in the study and their parents were selected for the questionnaire survey. For those who were screened positive, two psychiatry deputy chief physicians reviewed and diagnosed them according to the Diagnostic and Statistical Manual of Mental Disorders Fifth Edition (DSM-5) diagnostic criteria, and 40 children with ADHD who met the enrollment criteria and were matched in age and grade were selected, while one group of TD children matched in age, gender, and grade with the case group was selected as the control group. The mean age of the TD group was 10 ± 0.95, with 18 (45%) boys and 22 (55%) girls, and the mean age of the ADHD group was 9.9 ± 1.02, with 24 (65%) boys and 14 (35%) girls. The inclusion criteria were as follows: (1) age ≥9 years old, students in grades 3–5 of elementary school, to ensure that the participants could understand the test and had already comprehended the orthographic rules of Chinese characters; (2) Wechsler’s IQ test of more than 85; (3) no hearing or visual abnormality, and right-handedness; (4) the diagnostic criteria of ADHD: meeting the diagnostic criteria and exclusion criteria of ADHD in the DSM-5. Exclusion criteria: the influence of other disorders and environmental factors are excluded, such as the influence of organic brain disease, physical illness, physical disability, emotional disturbance, visual and auditory sensation, and unfavorable cultural stimuli; (5) exclude comorbidities such as reading disorders, learning disabilities, and oppositional defiant disorder.

All children participating in the experiment had their parents sign informed consent forms. The children themselves also gave their consent, and those who did not wish to continue participating could terminate the experiment at any time, with their data excluded from the analysis.

Ethics statement

The Affiliated Kangning Hospital of Ningbo University granted Ethical approval to carry out the study within its facilities (Ethical Application Ref: NBKNYY-2020-LC-50).

Materials

Chinese version of SNAP-IV

Three subscales were included: entries 1 to 9 were the inattention subscale, entries 10 to 18 were the hyperactivity/impulsivity subscale, and entries 19 to 26 were the oppositional defiance subscale, with each entry being rated on a four-point scale: 0, not at all; 1, somewhat; 2, quite a bit; and 3, very much. Symptoms were considered to be present if the symptom entry was scored as 2 or 3. Only the inattention subscale and the hyperactivity/impulsivity subscale were used for assessment in this study. Screening for ADHD was considered positive if six or more of the nine entries in the inattention subscale or six or more of the nine entries in the hyperactivity/impulsivity subscale were coded as being present (Xia, Shen & Zhang, 2015). The SNAP-IV diagnosis of ADHD had a sensitivity of 0.87 and a specificity of 0.79 (Zhou, Guo & Chen, 2013).

Go/No-go task

The experimental program was prepared using E-prime 3.0, and the stimulus material was presented in the center of the computer screen. The experimental procedure was as follows: 10 trials of practice experiments were required before each formal experiment, press the “F” key when you see the number “2” (Go stimulus) and do not press the key when you see the number “8” (No-go stimulus), and the formal experiment was started after ensuring that participants were clear about the experimental task. First, a 1,000 ms black gaze point “+” was presented, and then a 1000ms stimulus, “Go” stimulus and “No-go” stimulus were presented randomly. The “Go” stimulus and the “No-go” stimulus were presented randomly, and participants were required to respond within the presentation period. When the “Go” stimulus was presented, participants were required to press the “F” keys quickly and accurately, while when the “No-go” stimulus was presented, they did not need to press the “F” keys. The experiment was divided into 2 blocks of 200 trials, with a ratio of 4:1 between “Go” and “No-go” stimuli.

Morphological Awareness task

Referring to Gu et al. (2007) method of designing experimental stimulus materials, the experimental materials include 40 homographs and 40 homophones each. The experimental materials were all high-frequency words, balanced in terms of the number of strokes, word frequency and other factors, and they were all vocabulary required to be mastered in the Humanities Teaching Edition language textbooks for grades 1–3. The experimental program was prepared using E-prime 3.0, and the stimulus materials were presented in the center of a computer screen with a black screen and white font. The stimulus size was 3 cm × 6 cm, and the experimental procedure was as follows: eight trials of practice experiments (four trials for each phrase) were required before each formal experiment, and the materials used in the practice experiments were no longer used in the formal experiments, so as to ensure that participants were clear about the experimental task and then started the formal experiments. First, the black gaze point “+” was presented for 500 ms, and then the stimulus was presented for 1,500 ms. Participants were asked to determine as quickly and accurately as possible during the stimulus presentation period whether or not the target stimulus that had just appeared was a true word, and if it was a pseudo-word, to press the “1” key, whereas a true word did not need to press the key. If it is a pseudo-word, press the “1” key, while the true word does not need to press the key.

Data analysis

The EEG was acquired through 32 leads of Brain Products (BP) (EEG model: BrainAmp; Amplifier model: BrainAmp MR32). The recording electrodes were placed with reference to the international 10–20 standard lead system, and the reference electrodes were connected to both earlobes. The impedance of the electrodes was less than 5 KΩ, and the sampling rate was 500 hz. Data analysis was performed using Matlab 2021b. After removing bad segments, bandpass filtering was performed at 1–40 Hz and notch filtering at 49–51 Hz. Independent component analysis (ICA) was used to correct electrooculogram (EOG) artifacts, and after artifact correction, trials with amplitudes exceeding ± 100 µV were removed. ERPs were analyzed for 1,000 ms after the stimulus was presented, with a baseline of 200 ms prior to the stimulus. The ERPs waveform maps were superimposed according to the experimental task conditions, and finally the waveform maps of the ERPs of all participants with different conditions were averaged to obtain a total average map for each subject.

Statistical methods

Statistical analysis was performed using SPSS 26.0. Behavioral data (reaction time and accuracy rate) from the two groups of children during task performance, as well as waveform characteristics (amplitude and latency) of the N200 and N400 potentials in the Cz electrode regions, were analyzed. A 2 (condition: Go, No-go) ×2 (subject type: TD group, ADHD group) two-factor analysis of variance (ANOVA) was performed on the correct rate as well as the mean wave amplitude and latency of the N200 in the No-go-signal task. A 2 (condition: Homograph, Homophone) ×2 (subject type: TD group, ADHD group) two-factor ANOVA was conducted on the correct rate as well as the mean wave amplitude and latency of the N400 in the morphological awareness task, and the difference was considered statistically significant at p < 0.05

Results

In the Go/No-go task, an ANOVA on correctness showed a significant main effect of condition, F (1, 78) = 46.56, p < 0.001, ηp2 = 0.23, with correctness in the Go condition being significantly higher than correctness in the No-go condition. There was a significant main effect of subject type, F (1, 78) = 8.61, p = 0.004, ηp2 = 0.52, and a significant interaction between the two, F (1, 78) = 4.67, p = 0.032, ηp2 = 0.29. There was a significant difference between the TD and ADHD groups in the No-go task reaction time, No-go condition correctness, and the average amplitude of the waveforms in the No-go condition, p < 0.001, p = 0.009, p = 0.034 (Table 1).

Table 1 Comparison of differences between the two groups of subjects in the Go/No-go task.

Go/No-go task	TD group M (SD)	ADHD group M (SD)	F (1,78)	p	
Behavior					
Go Accuracy	98.58(2.60)	97.48(4.68)	1.63	0.205	
No-go Accuracy	92.00(8.27)	84.80(14.71)	7.15	0.009**	
Reaction time	540.2(85.3)	625.8(120.7)	15.72	0.000***	
ERP					
Go N200	−0.49(4.22)	.0.86 (3.08)	2.59	0.112	
No-go N200	−1.44(2.89)	−0.27(1.73)	4.68	0.034*	
Go Latency	299.80(16.52)	302.30(17.13)	0.43	0.514	
No-go Latency	301.70(17.26)	299.95(15.40)	−0.54	0.59	
Notes.

* p < 0.05.

** p < 0.01

*** p < 0.001.

In the morphological awareness task, an ANOVA on correctness revealed a significant main effect of condition, F (1, 78) = 6.68, p < 0.001, ηp2 = 0.06, which showed that correctness in the homograph condition was significantly higher than correctness in the homophone condition. The main effect of subject type was significant, F (1, 78) = 7.47, p = 0.007, ηp2 = 0.02, but the interaction between the two was not significant, F (1, 78) = 0.73, p = 0.004. The difference between the TD group and the ADHD group across experimental conditions was not significant in terms of correctness rate, mean amplitude of the wave, and latency period, but the two groups presented a significant difference in terms of time to response (Table 2).

Table 2 Comparison of differences in morphological awareness between the two groups of participants.

Morphological Awareness task	TD group M (SD)	ADHD group M (SD)	F (1,78)	p	
Behavior					
Homograph Accuracy	61.98(11.31)	58.31(12.50)	1.87	0.175	
Homophone Accuracy	57.11(12.68)	51.97(14.32)	2.81	0.097	
Homograph Reaction time	829.47(337.54)	960.43(237.76)	−4.03	0.00**	
Homophone Reaction time	845.52(376.78)	1,014.06(272.62)	4.64	0.00**	
ERP					
Homograph N400	−2.68(2.88)	−2.33(2.18)	0.36	0.552	
Homophone N400	−2.91(4.30)	−2.19(2.42)	0.84	0.363	
Homograph Latency	393.60(24.00)	397.45(22.82)	0.53	0.470	
Homophone Latency	394.55(33.14)	393.75(28.58)	0.01	0.909	
Notes.

** p < 0.01.

The ANOVA for mean N200 amplitude revealed a significant main effect of condition, F (1, 78) = 6.37, p = 0.013, ηp2 = 0.039, the No-go condition induced greater N200 amplitude than the Go condition (Figs. 1A; 1C). There was a significant main effect of subject type, F (1, 78) = 4.31, p = 0.040, ηp2 = 0.027, and he ADHD group exhibited lower N200 amplitudes compared to the TD group (Table 1). However, the interaction between condition and subject type was not significant, F (1,78) = 0.03, p = 0.858. ANOVA results showed that the difference between the N200 difference wave (No-go minus Go) in the TD group and that in the ADHD group was not significant, F (1,78) = 0.061, p = 0.806 (Fig. 1B).

Figure 1 Participants in the TD and ADHD groups at the CZ point of the Go/No-go Task.

(A) Go and No-go condition N200 waveforms; (B) waveforms of the N200 difference waveform; (C) N200 topographies of Go and No-go conditions.

The ANOVA for N400 mean amplitude revealed a nonsignificant main effect of condition, F (1, 78) = 1.117, p = 0.342, ηp2 = 0.011. there was a nonsignificant main effect of subject type, F (1, 78) = 1.222, p = 0.270, and a nonsignificant interaction between condition and subject type, F (1,78) = 0.816, p = 0.816 (Figs. 2A; 2C). ANOVA results showed that the difference between the N400 difference wave (homophones minus homographs) in the TD group and the ADHD group was not significant, F (1,78) = 0.344, p = 0.559 (Fig. 2B).

Figure 2 Participants in the TD and ADHD groups at the CZ point of the Morphological Awareness Task.

(A) N400 waveforms of the homograph and homophone conditions; (B) waveforms of the N400 difference wave; (C) N400 topographies of the homograph and homophone.

Discussion

The present study investigated whether children with ADHD have deficits in response inhibition and Chinese morphological awareness by combining behavioral and ERPs. Some researchers found that children with ADHD were significantly slower than TD children when responding to Go through a stop-signal task, and that children with ADHD had significantly lower N200 amplitude and longer latency than TD children, supporting the hypothesis that there is an inhibition deficit in children with ADHD (Senderecka et al., 2012). Some have compared response inhibition in children with ADHD and TD children via a controlled stop-signal task, showing that the mean response time of children with ADHD is significantly different from that of children with TD, presenting a slower response time (Janssen et al., 2015). Response inhibition has also been investigated in 5-year-old male children using a step-dynamic stop-signal paradigm, which showed that children with more attention-deficit hyperactivity disorder symptoms had a lower amplitude of the N200 (Berger et al., 2013). It has also been found that children with ADHD have prolonged N200 wave latency and reduced amplitude in the Go/No-go task, and that the right prefrontal N200 wave amplitude is reduced in the No-go trial (Anjana, Khaliq & Vaney, 2010; Pliszka, Liotti & Woldorff, 2000). The results of the present study showed that children with ADHD had longer reaction times in the Go condition and lower mean wave amplitudes in the No-go condition compared to TD children. It was also found that there was no significant difference in correct rates between the two groups in the Go condition, but there was a significant difference in correct rates and mean amplitude of responses between the two groups in the No-go condition, which were significantly lower in children with ADHD than in TD children, revealing the presence of impaired inhibitory abilities in children with ADHD. The absence of a difference in correct rates for the Go stimulus may be due to the fact that high-frequency stimuli arouse internal motivation in children with ADHD, whereas, under No-go stimuli, there is impaired executive ability, probably due to the presence of impulsive behavior in children with ADHD, who are unable to effectively inhibit stimuli that do not require a response, verifying the existence of a certain degree of response inhibition deficits in children with ADHD, which is basically in line with the results of previous studies (Wu et al., 2024; McGrath et al., 2011).

Research on language processing deficits in children with ADHD is predominantly conducted within English-language systems, with relatively few studies focusing on Chinese-language contexts. English, as a phonographic writing system, places phonological processing deficits at the core of dyslexia. In contrast, Chinese is an ideographic script, and its semantic processing model for reading follows a “form-semantic-phonological” sequence. Some scholars propose that Chinese reading involves not only phonological processing but also coexistent morphological awareness, suggesting that deficits in morphological awareness constitute the core deficit in Chinese dyslexia (Shu et al., 2006). However, research on whether children with ADHD exhibit deficits in morphological awareness remains controversial. Some researchers investigated reading fluency in four groups of children—TD group, RD group, ADHD group, and AD+ADHD group—using phonological awareness tests, orthographic tests, and morphological awareness tests. Their results showed that children with ADHD performed worse than the TD group on morphological awareness tasks (homophone morpheme judgment and homograph morpheme judgment), indicating morphological awareness deficits in children with ADHD (Li et al., 2023). Conversely, other scholars examined basic language processing deficits in four groups (TD, RD, ADHD, and ADHD+RD) using phonological awareness tests, rapid naming tests, and morphological awareness tests. They found no significant differences between children with ADHD and typically developing children in phonological awareness, rapid naming, or morphological awareness, suggesting that children with ADHD do not exhibit morphological awareness deficits (Zhang et al., 2016). Meanwhile an N400 study of shaped homophonic misspelled phrases in children with ADHD found no differences in N400 waveforms between children in the ADHD group and TD children (Liu et al., 2019). There is also the phenomenon of reduced N400 wave amplitude and prolonged latency found in children with ADHD by semantic congruence/incongruence task, revealing the existence of linguistic semantic deficits in children with ADHD (Diazábal Alecha, Guerrero-Gallo & Sánchez-Bisbal, 2006). Some scholars have found semantic processing deficits in RD children, showing elevated N400 amplitude, prolonged latency, and longer response times; this typical N400 effect suggests that RD children have difficulty with semantic integration in mid-processing (Wang et al., 2017). This study revealed no significant differences in the accuracy rates of responses between children with ADHD and TD children on two morphological awareness tests: the homophone morphological test and the homograph morphological test. However, significant differences were observed in reaction time, indicating lower information integration efficiency in children with ADHD. This may be associated with deficits in executive functions, particularly impairments in inhibitory control and response preparation. Furthermore, no significant differences were found in the latency, mean amplitude, or difference waves of the N400 component elicited during morphological awareness tasks between ADHD and TD children. This suggests that children with ADHD exhibit comparable abilities to TD children in extracting and integrating information from semantically unrelated homophone and homograph pairs during early semantic processing. Combined with prolonged reaction times, these findings indicate that ADHD-related deficits are concentrated in later-stage information processing (e.g., decision-making, response execution) rather than early semantic processing. Both groups showed relatively low accuracy rates on morphological awareness tests, which may be attributed to TD children’s familiarity with the vocabulary used in the experiment. Nevertheless, the integration of homophone and homograph elements likely impeded efficient information integration even for TD children.

Conclusions and Limitations

In summary, the results of this study show that children with ADHD have deficits in response inhibition but not in morphological awareness. They differ from TD children only in reaction time. This study contributes to the current exploration of Chinese semantic processing in children with ADHD. To date, few domestic or international studies have examined the semantic processing abilities of Chinese children with ADHD, and most of these studies have used English as the linguistic background. Although there are differences in the semantic processing patterns of Chinese and English, previous studies have not fully addressed the language processing abilities of Chinese children with ADHD.

However, the current study has obvious limitations. Only children with ADHD and TD children were included, and dyslexic children were not included for further differentiation and discrimination. The interaction between response inhibition and morphological awareness disorder was not discussed further, which is an issue to be explored in depth in future studies. Future studies related to children with ADHD can include data such as near-infrared, functional MRI, and eye movement as a new support point.

Additional Information and Declarations

Competing Interests

Author Contributions

Human Ethics

Data Availability

The authors declare there are no competing interests.

Fang Cheng conceived and designed the experiments, prepared figures and/or tables, and approved the final draft.

Xinhui Hu conceived and designed the experiments, analyzed the data, prepared figures and/or tables, and approved the final draft.

Yawen Chi analyzed the data, prepared figures and/or tables, and approved the final draft.

Jie Yang performed the experiments, prepared figures and/or tables, and approved the final draft.

Changzhou Hu performed the experiments, prepared figures and/or tables, and approved the final draft.

Beini Wang performed the experiments, authored or reviewed drafts of the article, and approved the final draft.

Jingjing Cui performed the experiments, authored or reviewed drafts of the article, and approved the final draft.

Taoping Wu performed the experiments, authored or reviewed drafts of the article, and approved the final draft.

Lixian Chen conceived and designed the experiments, performed the experiments, authored or reviewed drafts of the article, and approved the final draft.

Rong Wang analyzed the data, authored or reviewed drafts of the article, and approved the final draft.

The following information was supplied relating to ethical approvals (i.e., approving body and any reference numbers):

The Affiliated Kangning Hospital of Ningbo University granted Ethical approval to carry out the study within its facilities (Ethical Application Ref: NBKNYY-2020-LC-50).

The following information was supplied regarding data availability:

Data are available at Zenodo:

Cheng, F., Hu, X., Chi, Y., Yang, J., Hu, C., Wang, B., Cui, J., Wu, T., Chen, L., & Wang, R. (2025). Response inhibition and Morphological awareness in children with attention deficit hyperactivity disorder: evidence from behavior and ERPs [Data set]. Zenodo. https://doi.org/10.5281/zenodo.15549578.

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
