# Peer review of "Response inhibition and morphological awareness in children with attention deficit hyperactivity disorder: evidence from behavior and ERPs"

_PeerJ, doi:10.7717/peerj.19863_

## Round 0.1 · original submission · Major Revisions

All 3 reviewers have major comments which must be addressed adequately.

Reviewer 1 ·

Basic reporting

(1) The authors say multiple times throughout that response inhibition predicts (!) ADHD but I am not sure this statement is valid. I am giving the authors the benefit of the doubt here and put it under reporting. RI may be correlated with ADHD but I am not sure we can talk about prediction because (a) are you sure about the directionality of the causal chain? And (b) RI as measured in cogn. Tasks fluctuates and is not made to be reliable in the classical test theory sense or very stable over time
(2) Overall I have to say I was a bit confused as to ‘why’ specifically the morpheme awareness and SST performance should be related. It seems to me that the authors also assumed that they are linked to different ERPs. So why did the authors choose to measrure both together? From reading the extremely short introduction I am left confused as well. Sure both may be related to ADHD but why would it make sense to test them in the study together? What is the goal here?
(3) If the authors had specific predictions or hypotheses please state them. Also include directionality and be specific about the masures

If the authors correct these I would expect the manuscript to be significantly longer and include more references as well

Experimental design

(1) With regards to the sample: where there other exclusion criteria such as having normal or corrected to normal vision or hearing?
(2) How did the authors arrive at the sample size?
(3) When it comes to the SST: (a) how well does the task align with the consensus paper by Verbruggen et al., 2019, Elife? Some information is missing like the calculation of SSRT and how the SSD was adjusted (b) please include a figure about the trial sequence and the stimuli shown.
(4) Please report the characertics of your sample (N, mean age, gender distribution, etc)
(5) what template did you follow when constructingt he Morpheme test and how did you make sure it was suitable for children who are still in essence language learners?

Validity of the findings

(1) Given that the authors choose to do a study in children I must wonder about the motivational component as well. For example experimental attrituin and fatigue are especially problematic in children (more so in kids with ADHD). How did the authors deal with this? Can they rule this out? How do the results change over time? One solution, although more complicated could have been to use a gamified stop signal task to combat motivation loss. Also was the task order counterbalanced or not?
(2) Would the authors please visualize the results form the cognitive tasks? This would be nice for the reader.
(3) I am not sure the authors did calculate SSRT and if they did how? Also the authors should report SSD, GoRT, p(response,signal) and other measures as well do indicate SST performance. Also in the results the authors talk about “correctness” but I am not sure I understand what this means because there are at least 3 types of errors possible to calculate in the SST.
(4) This is about the sample but also about the validity of the findings: how many students did you actually measure? You are sometimes talking about 4 groups but that I never see the sample descriptions anywhere. You mention 40 ADHD people but does that mean you had 160 students? Or just 40? Given that I just don’t know I am not sure if I can trust the results or not. From the df’s in the analysis I assume you had 80 students? But then why talk about 4 groups?
(5) Given that there are no real hypothesis or theories stated I am not sure I can evaluate how well the conclusion fits the research gap (if there is one).

Reviewer 2 ·

Basic reporting

The English is clear and does not require revision. The references provided are relevant and up-to-date, providing sufficient background to understand the study and its rationale. Tables and Figures were also relevant and clear. I only had some problems accessing the raw data, despite the authors providing a link to their OSF account.

Experimental design

The experiment is perfectly designed and the tasks, materials, and sample are clearly described. The statistical analyses are also described in detail, but I have some suggestions. When reaction times or correct proportions are the dependent variables, ANOVAs are not the best possible analysis because neither reaction times nor hit proportions are normally distributed. Instead, I have some alternative to suggest. From my viewpoint, the optimal analysis is performing mixed linear analysis, with the condition and the group as fixed factors (exactly as in the ANOVAs performed) and the participant as a random variable. This way, the analysis accounts for intra-individual variability and does not require normality in the dependent variable. My second-best suggestion would be obtaining the mean reaction time for each individual and condition and using those means as the dependent variable. Furthermore, if there were a sufficient number of reaction times per individual and condition, I would suggest fitting an asymmetrical distribution, such as the exGaussian, to the data of each participant and condition, and then using the estimated parameters as dependent variables.
I would also suggest the authors to analyze the intra-individual variability in reaction times apart from the mean RTs.
Last, as a very minor comment: “…”Go” stimulus and “Stop” stimulus were presented randomly. The “Go” stimulus and the “Stop” stimulus were presented randomly, and subjects were required…” seems redundant to me.

Validity of the findings

Problems with statistical analyses notwithstanding, the results are correctly interpreted and thoroughly discussed with previous literature. Conclusions are also well built and are consistent with the results obtained. I have no comments in this regard.

Additional comments

Thank you for the opportunity to review this manuscript. The authors performed a stop-signal task and a morpheme awareness task, a task which asks for participants to discriminate between words and pseudo-words and react every time a pseudo-word appears, on children with and without ADHD. This research finds larger reaction times in children with ADHD compared with typically developed children, and the manuscript interprets this finding as evidence of impaired inhibitory control on children with ADHD.

I have some additional comments which I hope would help improving this manuscript:

1. “Attention deficit hyperactivity disorder (ADHD) is a neurodevelopmental disorder that occurs mostly in childhood and adolescence”. There is a growing body of evidence pointing out that ADHD persists through adulthood in many cases. Instead, I would say that ADHD is one of the most common neurodevelopmental disorders in children and adolescents.
2. “Some studies have shown that response inhibition is a major deficit in children with ADHD (Alexander, DeLong, & Strick, 1986)”. I would add at least one more recent study, such as Senderecka et al. (2012), which is already cited in the manuscript.
3. “…there is a speech processing disorder in children with ADHD”. Would “deficit” be a better alternative instead of “disorder”?
4. I would suggest the authors to use person-first language in all instances.
5. I would also suggest the authors to switch “normal children” for “typically developind children”, and “subjects” for “participants”.
6. In the reference list, the year in Alexander et al. (1986) should have parentheses.
7. The reference by Thomas et al. is not correctly ordered in the reference list.
8. In the references by McGrath et al. (2010) and Wang et al. (2017), the journal is written twice and the volume, issue and page numbers are missing.

Reviewer 3 ·

Basic reporting

1. English needs to be thoroughly reviewed. Many sentences are too long, and multiple sentences are combined as one. For example, the first sentence of the abstract and one sentence in the discussion, which stretched from line 233 and ended in 243 (10 lines!). Across all sections, it isn't easy to follow the intention of the authors regarding what they are discussing and the rationale or background of the statements.

2. In terms of writing, the introduction was written poorly and did not provide enough information about why you did this study. ERP and morpheme awareness were mentioned without much background information. Also, the introduction included some vague statements and lacking references for some parts. It is also too short to give readers a good grip on the rationale for the study.
Some examples in the introduction are listed below.
- The first page of the introduction (lines 37-38): what is the source for the domestic prevalence of 6.26%? On the same page (lines 28-43), define the response inhibition first and discuss the details. The definition is in the middle of the discussion on response inhibition.
- On the same page (lines 44-45), are there any references supporting this statement?
- Lexical processing and speech production may be significantly impaired in ADHD. But why morpheme awareness is important here? Provide the background information.

Experimental design

1. The main issue is the lack of confirmation that children with ADHD indeed have language impairment. Children with ADHD were screened through DSM-5 criteria and the Chinese version of SNAP-IV. But they do not have any language-related components. It is true that many ADHD individuals have a comorbid condition of certain language impairment, but it is not the defining symptom of ADHD according to DSM-5. DSM-5 has no diagnostic item for ADHD in the language area. Also, SNAP-IV does not have any items for language impairment. So, without an independent measure of language impairment in children tested in this study, it is spurious to claim that children with ADHD have no morpheme awareness issue. What if those children tested in this study have ADHD but no significant language impairment in the first place?

2. The authors stated that parental informed consents were obtained. What about the assent from children? Were there any children who resisted the study? Did anyone drop out of the study before finishing all the tasks?

3. Stop-Signal task description: the same sentences about key press were repeated. Why there are two keys (F and Q) for the “go” response? They also seemed to be weird locations on the keyboard to press. What were the “go” and “stop” stimuli? There was no information about what the "stimuli" were in this experiment.

4. Morpheme Awareness Task: please present the examples of Chinese characters used for the experiment. Also, it says “the experimental materials … balanced in terms of the number of strokes, …, and other factors.” What are the other factors? Be specific.
More critically, it is difficult to understand how 40 homophones and 40 homographs were presented on the screen. There must be two characters presented to make sure that participants can tell if they are homophones or homographs. But then, you asked participants to distinguish real and pseudo words. Did you include fake words that look like real words? It isn't easy to understand how the experiment went and what the stimuli were. Also, you asked participants to press the key “1” if they saw pseudo words and not to press any keys for real words. So, the response time and accuracy are only for the pseudo words. No responses were collected for real words. Is this correct? If so, table 2 shows the data only from the pseudo-word condition. But then, how do you know if the stimulus is homophones or homographs?

5. Please provide all demographic information and test statistics (tables).

6. ERP analysis: why is the Cz electrode analyzed only? Any reason for this? Also, please provide the details of the EEG equipment. It seems that you used an EEG system from Brain Product. But readers need more information about the system (electrode types, amplifier model, etc.).
Figures 3 and 6 showed topological maps of ERP activation if I am not mistaken. But the captions say it is the Cz point.

7. Please present results in a systematic way. You need sub-headings to make it clear what tests/variables are being presented there.

Validity of the findings

1. As pointed out above, your discussion on intact morpheme awareness in ADHD is not warranted or even supported by your findings. You need to have a clear indication that child participants indeed had language impairment, which could not be confirmed through the DSM-5 screening or SNAP-IV questionnaire.

Additional comments

No comments.

---

## Round 0.2 · Minor Revisions

I am pleased to accept this manuscript for publication pending a few more minor revisions as requested by Reviewer 3. Please address these specifically and make it clear in your response letter as to how these have been addressed in the manuscript, including where practical excerpts of the revised text.

Reviewer 1 ·

Basic reporting

no comment

Experimental design

no comment

Validity of the findings

no comment

Additional comments

The article has improved in clarity, and the authors have addressed all my concerns, although they did so in a very brief fashion in the manuscript. They also included more data tables and made it clear that it is not a stop signal but a go/no-go task.

Overall, the manuscript is straightforward, and I feel it is fine to publish.

Reviewer 2 ·

Basic reporting

No comment

Experimental design

No comment

Validity of the findings

No comment

Additional comments

The authors responded to our inquiries and comments in a perfect manner. I have no further comments.

Reviewer 3 ·

Basic reporting

The writing and reporting in the revised manuscript have improved from the original version. The authors responded well to my questions/comments. But their responses should be reflected in the revision, not just in their response to my comments. For example, add the specific EEG system information in the manuscript. Also, if the assent from the children was sought, this should be explicitly mentioned in the manuscript. Excluding children with dyslexia should also be mentioned explicitly in the list of exclusion criteria, not at the end of the manuscript.

Lines 87 -90. It says there are several studies connecting poor inhibition and poor comprehension. But there is only one study cited. This is an important part because the main logic for adopting morphological awareness in ADHD is due to the problem of inhibition.

I commented on writing in the original version of the manuscript. It has been improved, but please double-check the writing and punctuation again. For example, there is one sentence that spans 287 to 297 lines (10 lines!).

Experimental design

Thanks for adding the details of the test procedures and stimuli. See my comments in the “basic reporting” section.

Validity of the findings

I see that your idea here is to confirm impairment in morphological awareness among ADHD children because aberrant inhibition is key to ADHD and morphological awareness. But there is still a potential concern because not all children with ADHD have language impairment. As you stated in lines 77 to 82, morphological awareness is a core component of language processing. Not finding a difference in morphological awareness between typical children and children with ADHD is just because most participants with ADHD just happened to have no significant language impairment. Given that the connection between inhibition and morphological awareness is still controversial, as you stated, it is important to acknowledge this issue as a limitation clearly.

Additional comments

Please check the writing and punctuation. It still needs some work.

---

## Round 0.3 · accepted · Accept

I believe you have now addressed the remaining minor concerns and this paper is acceptable for publication with PeerJ.